**Impact of an educational program on earthquake awareness and preparedness in Nepal**
Shiba Subedi[1], György Hetényi[1] and Ross Shackleton[2]
[1]Institute of Earth Sciences, Faculty of Geosciences and Environment, University of Lausanne,
Switzerland
[2]Institute of Geography and Sustainability, Faculty of Geosciences and Environment, University
of Lausanne, Switzerland
**Keywords: School education, Hazard, Attitudes, Nepal, Science and education**
**ABSTRACT**
Scientific education of local communities is key to help reduce the risk associated with natural
disasters, such as earthquakes. Western Nepal has a history of major seismic events and is highly
prone to further earthquakes; however, the majority of the population is not aware about or
prepared for them. To increase earthquake awareness and improve preparedness, a seismology
education program was established at 22 schools in Nepal. In each school, educational activities
were performed by teaching earthquake related topics in classrooms, offering training to teachers
and through installing a low-cost seismometer network which supported both teaching and
awareness objectives. To test the effects of this program we conducted two surveys with school
children, one before and one after the initiation of the program, with several hundred participants
in each. The survey findings highlighted that educational activities implemented at schools are
effective in raising awareness levels of children, promoting broader social learning in the
community thus improving the adaptive capacities and preparedness for future earthquakes.
However, perceptions of risk did not change so much. The high and positive impact of program
on the students and the community is encouraging to continue and expand the program.

## INTRODUCTION

It is becoming increasingly important to educate people in the era of global change about environmental hazards to ensure they are well prepared to face the rising number of challenges. Education may play a central role for the risk management of natural hazards and help to reduce vulnerability and improve adaptability though allowing people to anticipate and prepare for hazards (Godschalk, 2003; IRGC, 2005).

Exact earthquake prediction is currently not possible, but responses to such events can be prepared for in advance to mitigate the effects they can have on society and human well-being (Turner, 1976). The impacts of earthquake disasters can be minimized by learning what to do before, during and after earthquakes, and by taking a variety of personal safety measures (Lehman & Taylor, 1987). Whether people prepare for future earthquakes or not can be significantly influenced by their education and their engagement on the topic (Tanaka, 2005). All-inclusive public awareness and education is fundamental to reducing causalities, personal injuries, and property damage from natural disasters (NRC, 1991; Torani et al., 2019). Researchers can contribute and play a key role in the education of society; not just to engage more people in research, but also to provide scientific explanations for natural hazards and related consequences to local communities and help to develop polices for mitigation of effects.

Earthquakes are the most common and deadliest natural hazard in Nepal with a long history of impacts in the country (Bollinger et al., 2016). Historical records indicate that many houses and temples in Nepal collapsed during the 1255 earthquake, and one third of the population including the King, Abhaya Malla, was killed. There are also records of an earthquake with a moment magnitude > 8 in 1505 (Ambraseys and Jackson, 2003) and indications that even larger earthquakes are plausible in the Himalayas (Stevens and Avouac, 2016). In 1934, during an earthquake (Fig. 1) with a moment magnitude (Mw) of 8.2 over 8'500 people lost their lives, 200'000 houses were severely damaged and more than 80'000 buildings completely collapsed (Dixit et al., 2013). The most recent major earthquake (Mw 7.8), in 2015, hit central Nepal resulting in about 9'000 causalities, and nearly 800'000 buildings were damaged or destroyed, leaving millions of people homeless. The resulting losses were equivalent to 50 % of total national GDP (Chaulagain et al., 2018). In addition, 19'000 classrooms were destroyed and 11'000 damaged (NPC, 2015b). It is suggested that if people had better awareness, preparations could have been more adequate and the negative impacts might have been lower (Hall & Theriot, 2016).

In Nepal, the National Seismological Center under the Department of Mines and Geology has been
conducting seismic monitoring since 1978. The Dept. of Education is responsible to develop
different educational activities across the nation, and the Dept. of Urban Development and
Building Construction has been working for building codes design and implementation. After the
2015 earthquake, the National Reconstruction Authority has been established and works for
reconstruction of buildings damaged during the Gorkha earthquake. Despite these efforts, the topic
of earthquakes is not included at any level of the official school curriculum in the Nepali education
system. However, recently the National Society for Earthquake and Technology (NSET) initiated
the Public-School Earthquake Safety Program in Nepal, but only in a few districts of the country
(Dixit et al., 2014). This program focuses mainly on the retrofitting of school buildings to restore
and minimize future damage following the 2015 earthquake; however, educational efforts are still
very limited.
Following the devastating 2015 Gorkha event, and considering the history of major earthquakes
and the likelihood of many more as well as poor educational effort on the topic, we initiated and
implemented a seismology education program in schools in Western Nepal (Fig. 1; Subedi et al.,
2020) including the area affected by the 2015 earthquake and expanding towards the West (Fig.
2). The aim of the program is to increase the earthquake awareness levels in Nepal, starting from
the schools, with the hope that this knowledge will be spread into the community through social
learning, and partly through the establishment of a low-cost seismic network (Figs. 1, 3).  In this
study, the effects of the education program for earthquake awareness and preparedness are
evaluated. The evaluation was performed by collecting data from students through two surveys,
one before and one after the initiation of the education program.
**METHODS**
The data for this study were collected using two questionnaire surveys on paper, conducted in
Nepali language: in 2018, before the initiation of the education program, and in 2020, nearly a year
after the full implementation of the program.
Before the initiation of the education program, we undertook fieldwork to help inform our strategy
and the educational materials, and to ensure the education program was well adapted to the Nepali
education system. In 2018, during the first visit, we talked with the school leaders about the
program and its benefits, and gave sample lectures (ca. 1-2 hours including questions) to students
between the ages of 14-16, providing key information on earthquakes. Before the sample lecture
and in each school, students were requested to fill in a paper questionnaire survey on earthquake
related questions. In special lectures we also taught students how to prepare before an earthquake,
how to save lives during an earthquake, and what to do after an earthquake. We also provided a
flyer containing detailed information and pictures (Fig. 4), of which we distributed 500 copies.
Similarly, we designed a sticker to remind people about earthquake hazards (Fig. 3), and
distributed this to students and teachers (3'000 so far).
In April-May 2019, during the second visit, the program was fully implemented with the
installation of an educational, low-cost seismometer in every school. The seismometer's record is
displayed on a computer, which is easily accessible to students in their physics class, or through
an online application. During the visit, we also identified the open place near the school where
students should meet in case of earthquake and installed an Emergency Meeting Point sign in
Nepali. To increase the efficiency of the learning and to ensure long-term uptake, we organized a
2-day workshop for nearly 100 school teachers, which was very well received. The full details of
the program are documented in an earlier paper (Subedi et al., 2020) and all the material is
accessible on the program website (www.seismoschoolnp.org).

In this article, we focus on evaluating the efficiency of our program in terms of knowledge and
behavior change of students related to earthquakes. Out of 22 schools participating in the program,
15 schools were chosen for the survey, covering a range of socio-economical contexts. Students
for the surveys were selected randomly from grades 9 and 10, representing the 14-16-year-old age
group. The total number of responses collected was 318 in 2018 and 480 in 2020, respectively.
For logistical reasons, some responses in the pre- and post-survey (27 %) came from different
schools, but this is not expected to affect the results as they were independent samples. While the
first set of students surveyed had no earthquake education whatsoever, those who filled out the
second survey were exposed to information and lectures frequently about earthquakes from the
teachers who were trained in our program.
When the exact same question was asked before and after our program's implementation, we
quantify the change using $\chi^2$ test analysis. In doing so, our null hypothesis ($H_O$) is that our program
had no effect on the students. If this null hypothesis is unconfirmed (i.e., the $\chi^2$ value is above the
threshold for the corresponding number of possible answers, and the respective p-value is below
5%), then we interpret that the program had an effect on the students as their answers show a clear,
statistically significant change. The complete set of questionnaires are available in the
Supplementary materials file.



**RESULTS**
The first measurement of this study, performed in the 2018 survey, was about the experience of
the 2015 Gorkha earthquake. The majority of respondents, 94 %, felt the shaking. As the
earthquake was on Saturday, schools were closed and students were at home; 71 % of students
answered that they ran out of a building, and only 15 % hid under a table, 8 % did not know what
to do, 3% stood next to the wall or the doorframe, 3% had other reaction.

**Knowledge about the causes and possibility of earthquakes in Nepal**
Before the implementation of the program, 7 % students believed that earthquakes were caused by
a moving fish carrying the Earth (a Hindu belief and myth). However, 64 % still chose the correct
scientific answer: plate tectonics. The majority of students, 84 %, chose the "plate tectonics"
answer in 2020, and the percentage of responses relating to the cultural/religious reasons dropped
to 2 % (Fig. 5).
Regarding the probability of a future earthquake greater than in 2015, more students knew that
such an earthquake in their region was quite likely after the education program (Fig. 6a). At the
same time, there was a clear drop in the number of responses for very unlikely (17 % in 2018 to 5
% in 2020) and a slight drop in the percentage answering that a future great earthquake is
impossible.
Relating to the effects of a Mw > 8 earthquake, after the program, the answer *I could die* has
increased by a factor of 1.8, and all other answers (*I could be buried alive*, *I could get hurt*, *I could*
*lose friend* and *My home could collapse*) are increased by a factor of at least 1.3 compared to 2018
(Fig. 6b; multiple answers were possible).
In 2018, 31 % students answered they know when an earthquake will occur, which is reduced to
11 % in 2020. The answer itself is not true, and this mis-information could drive people to
incorrectly prepare for or act during an earthquake. While our efforts clearly decreased this mis-
conception among the students, we could not yet reach each and every student to teach them about
the unpredictability of earthquakes. The students answer agreeing on the impossibility of
preventing an earthquake was 86 % in 2020, showing an   absolute increase of 18 % from 2018.
This question also shows that by 2020, more than double of the respondents have participated in
disaster risk education training compared to 2018 (Fig. 7).

**Knowledge and perceptions about how to behave during and after an earthquake**
Three quarters (75 %) of students in 2020 responded that their family knows what to do and where
to go during an earthquake, an increase of 55 % from 2018. Only 37 % of students in 2020 believed
that their home could resist a large earthquake. For comparison, 65 % students were scared and 22
% panicked during the Gorkha earthquake in 2015 (10 % had calm reactions, 3 % did not care)
according to answers in 2018.
In 2018, 62 % respondents didn't know that they should not call others after an earthquake to leave
the phone lines available for rescue operation, but in 2020 nearly 80 % students knew this useful
practical point (Fig. 8).
After the implementation of our program, 65 % of the students believed that they can survive if a
large earthquake occurred at night, whereas 43 % felt they could survive in 2018. This information
reflects more confidence of students as they become familiar with earthquake topics and heard
more information about them.
In 2020, 93 % of children knew that during an earthquake, the majority of injuries and deaths are
caused by people being hit by objects, through the collapse of constructions; the proportion of
people not knowing this dropped by 2/3 after the educational program was implemented. More
than 2/3 of the students in 2020 were aware about the additional hazards, such as fires, landslides
and floods that can be triggered by an earthquake. There is a 7% decrease for this answer since the
2018 survey, but as students who claimed partial knowledge increased by 7 % as well, a net change
in knowledge is not really perceptible on this point.
The proportion of students who regularly discuss earthquake related topics within their families
has increased by 18 % (absolute increase; see Table 1). This shows that the education program at
schools has led to widespread social learning within communities. This is reinforced by the finding
that nearly all students (98 %) are interested to learn more about earthquakes in detail, which will
aid communities towards better earthquake preparedness in the long run.

**Earthquake preparedness and adaptation**
In 2018, 36 % of students perceived that to remain alive during an earthquake depends on luck,
while this number has decreased by a relative 60 % after our program started and is a concern for
only 21 % of students (Fig. 9). All possible answers regarding adaptation options to earthquakes
record an increase from 2018 to 2020 (Fig. 11). The majority (72 %) of respondents answered that
they are aware of the shelter areas and open spaces where they can go in case of an earthquake.
The same proportion of people are aware of evacuation areas in 2020, but the increase here is much
more important (from 38 to 69 %), potentially thanks to the Nepali Emergency Meeting Point signs
we installed in schools. The information about which governmental authority to contact after an
earthquake is relatively low, but has increased by 10 % (absolute). Information about earthquake
prone areas and the reception of knowledge on earthquake disaster adaptation have increased by
the factor of 2.5, from 12 % in 2018 to 31 % in 2020 after the education program.
The relatively small number of respondents who claimed that the government will provide help
after an earthquake increased by a factor of almost 3: from 8 % in 2018 to 23 % in 2020. This
percentage is not yet sufficient in general, but the improvement following our program's
implementation is noteworthy. Moreover, the level of confidence in the government's
reconstruction activities has also grown, from 13 to 30 %, which is a good sign and shows
increasing level of trust. In 2020, 68 % of the respondents knew about the importance of talking
about earthquakes with neighbours, friends and colleagues, a nearly two-fold increase in two years.
Furthermore, we found that all students discussed their new knowledge and learning about
earthquakes with the people around them in the community. Ninety-one percent of the students
talk at least with some people in the community, only 9 % discuss this with their parents only, and
there is no student who had not had a discussion in her/his surrounding (Fig. 10).

**Perception of risk**
More than 60 % of the answers showed that students considered the level of seismic risk in their
city as medium, which means their risk perception is underestimated with respect to the actual
seismic risk level in the region (Stevens et al., 2018). Only every 6[th] person claims to perceive high
risk, which is clearly less frequent than people declaring low risk. As opposed to our expectation,
there is very little change in the level of risk perception in the group of students from 2018 to 2020:
the medium risk level group is the same, and there is minor change in low and high-risk level
groups (Fig. 12).  This result is a surprise, especially when compared to the 72 % of responses in
2020 who believe that there is more than 70 % chance of experiencing an earthquake larger than
the 2015 Gorkha earthquake in their life (Fig. 6a).

**Project acceptance and future education**
To measure the program's acceptance level, some questions regarding the program itself were also
included in the 2020 questionnaire. It is found that 91 % of the students know that a seismometer
is installed in their school for earthquake education purposes. A total of 61 % of the students have
observed waveforms recorded by the seismometer, either at the school computer (39 %), on the
teacher's mobile phone (18 %) or/and on their parents' or own mobile phone (8-8 %). Furthermore,
85 % of the students answered that teachers teach about earthquakes in the classroom regularly
(weekly, monthly, on demand, and/or following an earthquake). In 2020, 99 % of the students
expressed that they like the earthquake information we have provided them. Regarding future
plans, almost all students are very much (69 %) or simply (29 %) interested to learn about
earthquakes by inserting the theme in the official curriculum, which can be instituted by the Local,
Provincial and Federal Government of Nepal as they have all have some field of possible action.
Hence, our program and the methods we use for teaching about earthquakes are well accepted.

**Statistics**
All questions except the last (Question 12 in Table 1, level of interest to learn is 98% in both
surveys) record a clear change in the pattern of answers given following our program's
implementation (see Supplementary Table 1). The biggest statistical change was seen for Question
6 (avoid post-earthquake use of mobile communications) suggesting a big increase in knowledge
and a very new information. Each question (excluding those with multiple choice answers) and
their corresponding $\chi^2$ and p-values are reported in the Supplementary Table 1.


## DISCUSSION

### Have earthquake awareness levels increased?

As a result of the novel school-based education program, themes related to earthquakes are more familiar to the students now than in the past, and their awareness level have increased since the program was initiated. Students know more about the earthquake phenomena and have changed their behavior to better prepare and adapt to forthcoming earthquakes. Earthquake related knowledge learnt by students at schools has also reached across the broader community, though social learning processes (Reed et al. 2009).

### Why have the awareness levels increased?

Beyond the prescribed school education, our program has provided an opportunity for informal and free-choice education forms, in which people can learn about topics outside of formal educational settings, which has been well supported by enthusiastic teachers (Falk & Dierking, 2002). This form of social learning enables an increase in knowledge, and through further communication with others, it spreads knowledge in communities, which may lead to changes in attitudes, behavior, and building of trust in society (Reed et al., 2010). This method is widely applied for the study of natural hazards and its management (e.g., Brody, 2003; O'Keefe et al., 2010). During our program's implementation, despite being in contact only with the school children, the knowledge has spread much more widely in local communities through social learning, thus reaching and impacting the original and intended target group.

People's behavior can also be developed through education. The idea is that if people are made knowledgeable of earthquakes, they are more likely to adopt and perform behaviors that will increase their earthquake awareness and preparedness (Hungerford and Volk, 1990). This has similarly been shown for other environmental issues like invasive species, where campaigns building knowledge and awareness changed behaviors therefore minimizing risk (e.g. Cole et al. 2019).

As a result of our educational program, earthquake related knowledge has increased and the behavior to cope with earthquakes has also changed. Despite this, the earthquake risk perception of students has not fully changed yet. Our results show that a realistic and appropriate distribution of earthquake related knowledge and increased awareness level are not (or not yet) sufficient to influence the perception of risk. Perceptions are a complex phenomenon and can take a long time

to change (De Dominicis et al., 2015; Estévez et al., 2015; Cole et al., 2019; Shackleton et al.,
2019). Education and awareness raising is the key factor for changing long-term risk perceptions
– although programs need to be well tailored to appropriate audiences (Lee et al., 2015). Although,
some studies discuss that increased knowledge does not always relate to increased risk perceptions,
and increasing perceived risk does not necessarily result in the reduction of risk behavior (e.g.
Noroozinejad, 2013; Petros, 2014). In addition, knowing more of a given topic makes people more
certain, self-confident, which may lead to underestimate the related risk (e.g. Stringer, 2004).
Moreover, increased knowledge and behavior to adapt and feel more secure during an earthquake
should reduce the fear of associated risk and therefore reduce the risk perception. The limited
change in risk perception in this study may be due to better knowledge of the hazard and how to
mitigate it (Ndugwa Kabwama and Berg-Beckhoff, 2015).
Hence, how people perceive risk is not necessarily related to the actual risk. We cannot draw a
definitive conclusion as the related knowledge can contribute to the amplification or the
attenuation of the related risk; as such, it could be one of the potential reasons for the low risk
perception of people having more knowledge (Reintjes, 2016). Risk perception is thus important
for preventative actions, but risk perceptions are often biased (Weinstein, 1988). It could be that
more time is needed to change students' risk perceptions, and it is also likely that there are other
factors such as economic status, gender, age group, location of home in city, etc. that may influence
the level of risk perception of people. A repeated survey in the same age category in a few years'
time may give an answer to this question. We suggest that further monitoring and adaptation of
the education system might be needed to better link awareness raising, behavior change and risk
perception change.


**Further action needed**
Since other sources of information, such as newspapers and television, are not easily available to
people in the Nepali countryside, we believe that the school is the best platform to transfer
knowledge to the community. The proper education at school reaches deep across the families and
into the community, and the discussions in those circles are essential to prepare the whole society
for future earthquakes. The proportion of students who regularly discuss earthquake related topics
within their families has increased by 18 % (absolute increase; see Table 1). This shows that the
education program at schools has led to widespread social learning within communities, and
possibly beyond our program's current area. We therefore, advocate for a continuity of this
program and to get education about environmental hazards more deeply embedded in the Nepali
education system.

Although this program has increased the earthquake awareness level among students and the
broader community in the program area, it is alone not sufficient for seismic risk reduction. Further
monitoring and adaptation of the program to promote changes in risk perception and improved
learning is advised. Education will help communities to prepare for future earthquakes, but the
local, national and regional governments are responsible for the rescue, support and reconstruction
operations in the case of a severe earthquake and well as developing and implanting policy to
mitigate against threats. People's situation after an earthquake depends on how well they are
prepared for the event, so developing policy, for example, on construction quality depending on
expected shaking intensities is advised. Since the shaking level of an earthquake cannot be
controlled, the impact of an earthquake on the community is strongly dependent on the actions
taken by the government for its preparedness, such as education (so far our program's effort) as
well as, for example, a suitable, locally calibrated and enforced building code. For both aspects,
the provincial governments could overtake some of the efforts drawing on our bottom-up
approach, and adapt them to continue earthquake education in schools, which is an efficient way
to make earthquake safer communities. In parallel, local initiatives are encouraged to strengthen
these efforts.

**CONCLUSIONS**
The Seismology at School in Nepal program has been successfully implemented and achieved the
aim of raising earthquake awareness and preparedness by educating students in their schools. The
program itself and the methods we used for teaching about earthquakes and demonstrating with
low-cost seismometers are well accepted by students and teachers. The new knowledge learned by
the students at school reaches their parents and is transferred into the local community. The results
we observed through two surveys, before and after initiation of the education program, are
measurable, statistically significant and with positive changes for earthquake related knowledge
and preparedness level, but not (yet) for the perception of the related risk. A high and positive
impact of the program on the students and their communities is encouraging for the continuation
and expansion of the program in the region. Governmental institutions are encouraged to build on
this experience as well as develop further policy to mitigate the risk of future earthquakes in Nepal.

**ACKNOWLEDGEMENTS**
We greatly acknowledge students, school teachers and principals from the school participating in
the program. We are very thankful to people who helped carrying out the surveys.  We highly
appreciate the American Geophysical Union for their AGU-Celebrate-100 grant support which
allowed us to invite Nepali teachers to the workshop.  We greatly acknowledge the Institute of
Earth Sciences and the Faculty of Geosciences and Environment at the University of Lausanne for
hosting Shiba Subedi as a doctoral student, and for their support for instrumentation. The funding
from Federal Commission for Scholarships for Foreign Students, Switzerland, for Shiba Subedi's
PhD thesis is well acknowledged. We warmly thank Anne Sauron, Peter Loader and Paul Denton
for valuable suggestions and useful discussions. We are also thankful to Mrs. Apsara Pokhrel for
translation and typesetting of the survey questionnaire in Nepali language.

Figure 1: Map of Nepal, with the locations of schools participating in the Seismology at School in
Nepal program. Background represents population density data (CIESIN and CIAT, 2005). The
Main Frontal Thrust (MFT), the surface trace of the fault underlying most of Nepal and hosting all
great earthquakes in the region, is indicated in red solid line. Three colored segments represent the
rupture extent of the corresponding major and great earthquakes with moment magnitude (Mw) as
indicated (after Bollinger et al., 2016). For the 2015 Gorkha earthquake the rupture area is also
plotted (blue contour). Letters P and K refer to cities Pokhara and Kathmandu, respectively,
marked with black circles.

Figure 2: Students gathered at the morning assembly in the *Shree Himalaya Secondary School,*
*Barpak, Gorkha* district. The school building was damaged during the 2015 earthquake and
students were in temporary shelters. The construction of the new building is visible at the top of
the picture. (Photo: S. Subedi, in May 2018, with permission of the school).

Figure 3: Left: The Raspberry Shake 1D low-cost seismometer, installed in 22 schools across
Central Nepal (Fig. 1). Right: Earthquake awareness sticker, as a reminder, in English and Nepali
language (artwork of M. Dessimoz). The sticker image is available for download from our
program's webpage: www.seismoschoolnp.org.

Figure 4: Educational flyer in Nepali language on what to do before, during and after an
earthquake. The flyer has been translated and adapted from an English version, compiled by and
available from the CPPS earthquake education centre in Sion, Switzerland (www.cpps-vs.ch). The
Nepali flyer is available for download from our program's webpage: www.seismoschoolnp.org.

Figure 5: Student opinions on what causes earthquakes (Q1), before and after the initiation of our
education program. ($\chi 2 = 78.15$, p-value = < .00001, the change is significant).

Figure 6: *(a)* Student views on how likely the occurrence of a next earthquake bigger than the 2015
Gorkha earthquake is (Q3), before and after the initiation of our education program. ($\chi 2 = 43.59$,
p-value = < .00001, the change is significant). *(b)* Student answer on the outcome of a potential
Mw>8 earthquake in Nepal (Q2), before and after the initiation of our education program.
*Multiple answers were possible.

Figure 7: Students' personal knowledge about earthquakes (Q13), before and after the initiation of
our education program. *Multiple answers were possible.

Figure 8: Student's knowledge on the recommendation to avoid making phone calls after an
earthquake to leave lines available for rescue operations (Q6), before and after the initiation of our
education program. ($\chi 2 = 138.72$, p-value = < .00001, the change is significant).

Figure 9: Student's own opinion on earthquake preparedness (Q14), before and after the initiation
of our education program. *Multiple answers were possible.

Figure 10: Student activities to transfer the knowledge to the community (question e), after
initiation of our education program.

Figure 11: Student ideas about earthquake adaptation (Q15), before and after the initiation of our
education program. *Multiple answers were possible.

Figure 12: Students' perception of the level of seismic risk in their respective location (Q10),
before and after the initiation of our education program. ($\chi 2 = 6.33$, p-value = 0.042, the change is
slightly above significant level).






| No | Question | Answer in 2020 survey | | | Answer in 2018 survey | | |
|---|---|---|---|---|---|---|---|
| | | Yes | Partially | No | Yes | Partially | No |
| Q7 | If a large earthquake occurred at night, could you save yourself? | 65% | - | 35% | 43% | - | 57% |
| Q8 | Do you know that the majority of injuries that occur in earthquakes are caused by people being hit by or stumbling over fallen objects? | 93% | - | 7% | 76% | - | 24% |
| Q9 | Do you know that earthquakes can make additional damage such as fire, landslides and floods? | 68% | 21% | 11% | 75% | 14% | 11% |
| Q11 | The preparedness for a major earthquake is the most important thing. Are you | 71% | - | 29% | 53% | - | 47% |

| | | | | | | | |
|---|---|---|---|---|---|---|---|
| | regularly discussing this topic with your family? | | | | | | |
| Q12 | Are you interested to know more about earthquakes and its preparedness in details? | 98% | - | 2% | 98% | - | 2% |

Table 1: Questions and respective answers about earthquake preparedness among students who participated in the surveys, before and after our education program was initiated in Central Nepal. Respective statistical indicators are reported in Supplementary Table 1.

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

**CONFLICT OF INTEREST AND ETHICS**
The authors declare that the research was conducted in the absence of any commercial or financial
relationships that could be construed as a potential conflict of interest. The authors declare that an
ethical approval was not required as per local legislation. The authors declare that they have no
conflict of interest.

**AUTHOR CONTRIBUTIONS**
The project concept and implementation details were developed by S.S. and G.H. Most of the
fieldwork was carried out by S.S. with some help by G.H. The preparation of the manuscript,
figures, tables and the calculations were done by S.S. and guided and verified by G.H and R.S. All
authors discussed the results, and contributed to the final manuscript.

**SUPPLEMENTARY MATERIAL**
The Supplementary Material for this article can be found in supplementary material file.

**DATA AVAILABILITY STATEMENT**
The datasets used for this study can be available on request to corresponding author.

Figure 1

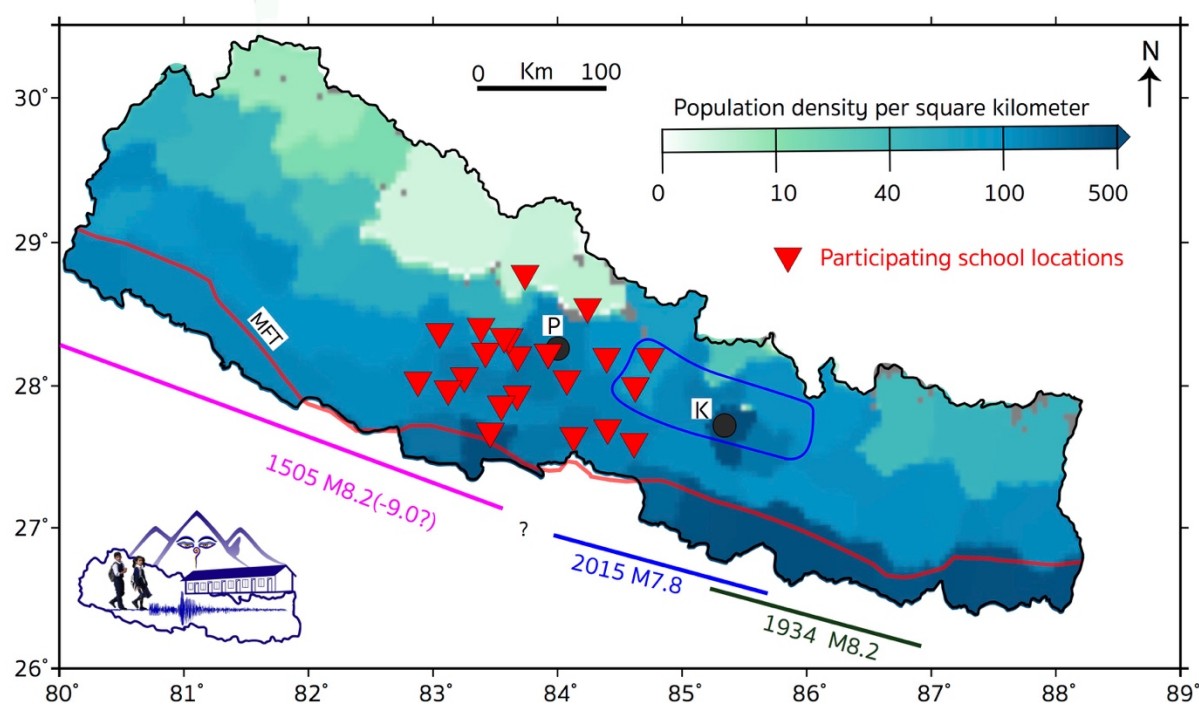


Figure 2

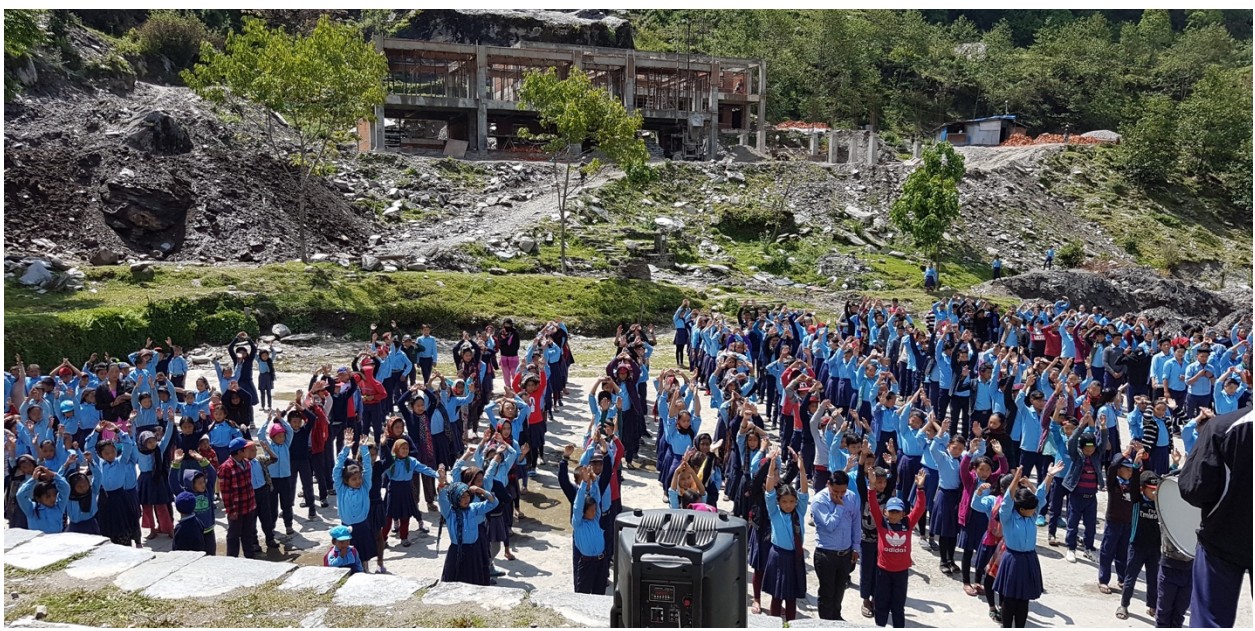


Figure 3

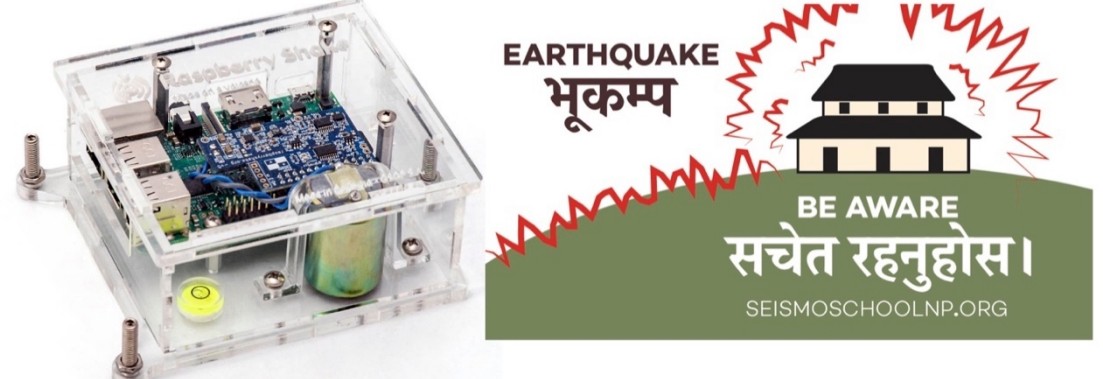
























Figure 4

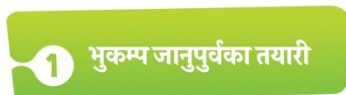 **१ भुकम्प जानुपुर्वका तयारी**

### सुरक्षित ठाउँ पत्ता लगाउनु

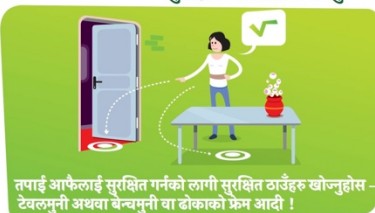

तपाई आफैलाई सुरक्षित गर्नको लागी सुरक्षित ठाउँहरु खोज्नुहोस् – टेवलमुनी अथवा बेन्चमुनी वा ढोकाको फ्रेम आदी !

### वरीपरी हेर्नुहोस्

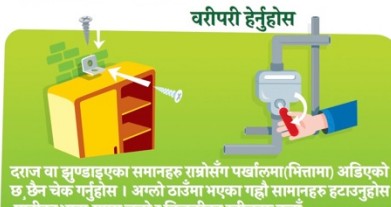

दराज वा झुण्डाइएका समानहरु राम्रोसँग पर्खालमा(भित्तामा) अडिएको छ छैन चेक गर्नुहोस् । अल्लो ठाउँमा भएका गह्रौ सामानहरु हटाउनुहोस् । पानीका भाडा, ग्यास चुलो र बिजुलीका स्वीचहरु कहाँ छन् याद गर्नुहोस् ।

### अत्यावश्यक सामाग्रीको तयारी

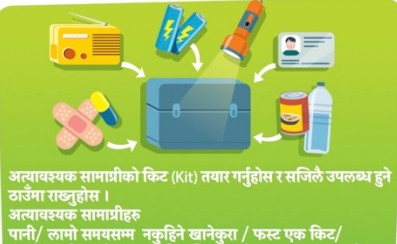

अत्यावश्यक सामाग्रीको किट (Kit) तयार गर्नुहोस् र सजिले उपलब्ध हुने ठाउँमा राख्नुहोस् ।
अत्यावश्यक सामाग्रीहरु
पानी/ लामो समयसम्म  नकुहिने खानेकुरा / फस्ट एक किट/ सानो ध्याट्री/ टर्चलाइट ब्याट्रि सहित/ तातो कपडाहरु/ब्ल्याङकेट/ आफ्नो परिचय दिने कागजको प्रतिलिपी/ केही पैसा आदी ।

### आफैले अभ्यास गर्नुहोस्/ तालीम लिनुहोस्

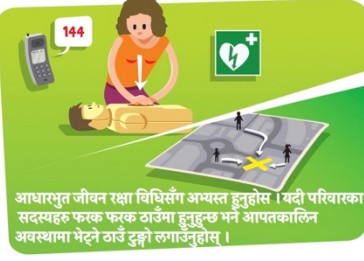

आधारभुत जीवन रक्षा विधिसँग अभ्यस्त हुनुहोस् । यदी परिवाका सदस्यहरु फरक फरक ठाउँमा हुनुहुन्छ भने आपतकालिन अवस्थामा भेट्ने ठाउँ टुङ्गो लगाउनुहोस ।

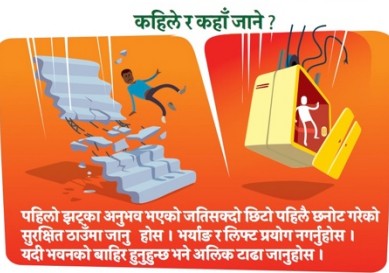 **२ भुकम्प जाँदै गर्दा**

### कहिले र कहाँ जाने ?

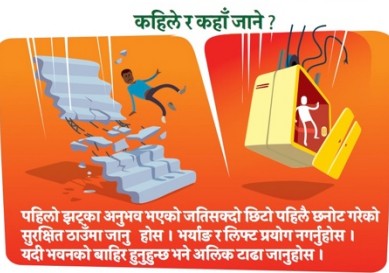

पहिलो झट्का अनुभव भएको जतिसक्दो छिटो पहिले छनोट गरेको सुरक्षित ठाउँमा जानु  होस । भर्याङ र लिफ्ट प्रयोग नगर्नुहोस् । यदी भवनको बाहिर हुनुहुन्छ भने अलिक टाढा जानुहोस ।

### आश्रयस्थल पत्ता लगाउनु

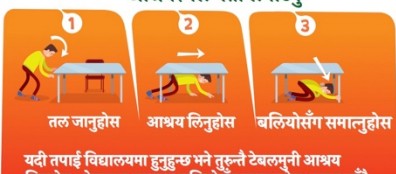

**तल जानुहोस्**   **आश्रय लिनुहोस्**   **बलियोसँग समाल्नुहोस**

यदी तपाई विधालयमा हुनुहुन्छ भने तुरुन्तै टेबलमुनी आश्रय लिनुहोस् । टेबलका खुट्टाहरु बलियोसँग समाल्नुहोस,भुकम्प जाँदै गर्दा टेबलहरु सर्न सक्छन ।

### भवन बाहिरको जोखिम

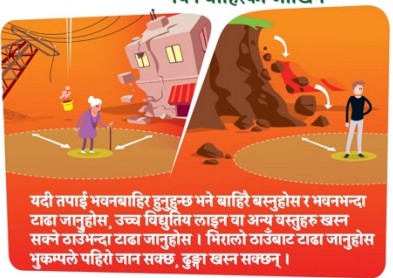

यदी तपाई भवनबाहिर हुनुहुन्छ भने बाहिरै बस्नुहोस र भवनभन्दा टाढा जानुहोस्, उच्च विद्युतिय लाइन वा अन्य वस्तुहरु खस्न सक्ने ठाउँभन्दा टाढा जानुहोस् । भिरालो ठाउँबाट टाढा जानुहोस भुकम्पले पहिरो जान सक्छ, ढुड्गा खस्न सक्छन ।

### कारभित्र/बसभित्र

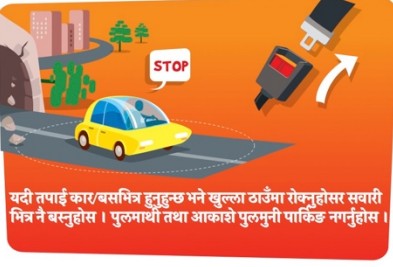

यदी तपाई कार/बसभित्र हुनुहुन्छ भने खुल्ला ठाउँमा रोक्नुहोसर सवारी भित्र नै बस्नुहोस । पुलमाथी तथा आकाशे पुलमुनी पार्किड नगर्नुहोस ।

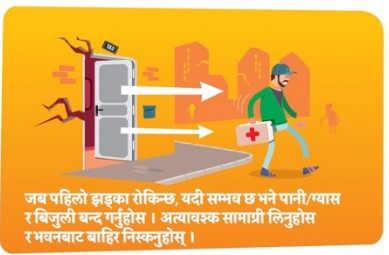 **३ भुकम्प पछाडी सतर्क रहनुहोस्**

### भुकम्पको झड्का सकिदा बित्तिकै

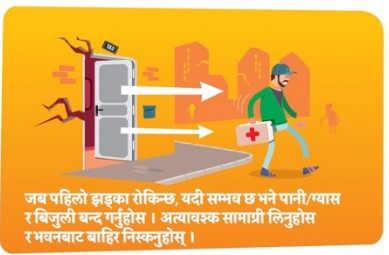

जब पहिलो झड्का रोकिन्छ, यदी सम्भव छ भने पानी/ग्यास र बिजुली बन्द गर्नुहोस् । अत्यावश्यक सामाग्री लिनुहोस र भवनबाट बाहिर निस्क्नुहोस् ।

### सावधानीपूर्वक बस्नुहोस

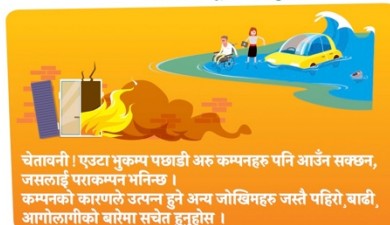

चेतावनी ! एउटा भुकम्प पछाडी अरु कम्पनहरु पनि आउन सक्छन, जसलाई पराकम्पन भनिन्छ ।
कम्पनको कारणले उत्पन्न हुने अन्य जोखिमहरु जस्तै पहिरो, बाढी, आगोलागीको बारेमा सचेत हुनुहोस ।

### मेडिकल केयरको सुनिश्चित गर्नुहोस्।

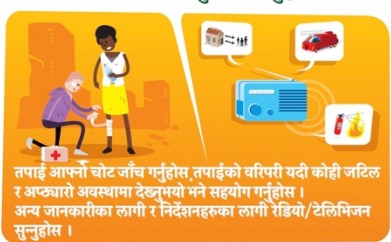

तपाई आफ्नो चोट जाँच गर्नुहोस्,तपाईको वरिपरी यदी कोही जटिल र अप्ठ्यारो अवस्थामा देख्नुभयो भने सहयोग गर्नुहोस । अन्य जानकारीका लागी र निर्देशनहरुका लागी रेडियो/टेलीभिजन सुन्नुहोस ।

### अत्यावश्यक सेवाहरु

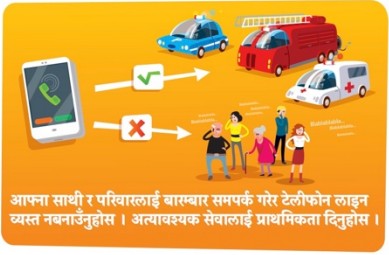

आफ्ना साथी र परिवारलाई बारम्बार सम्पर्क गरेर टेलीफोन लाइन व्यस्त नबनाउनुहोस् । अत्यावश्यक सेवालाई प्राथमिकता दिनुहोस ।




Figure 5

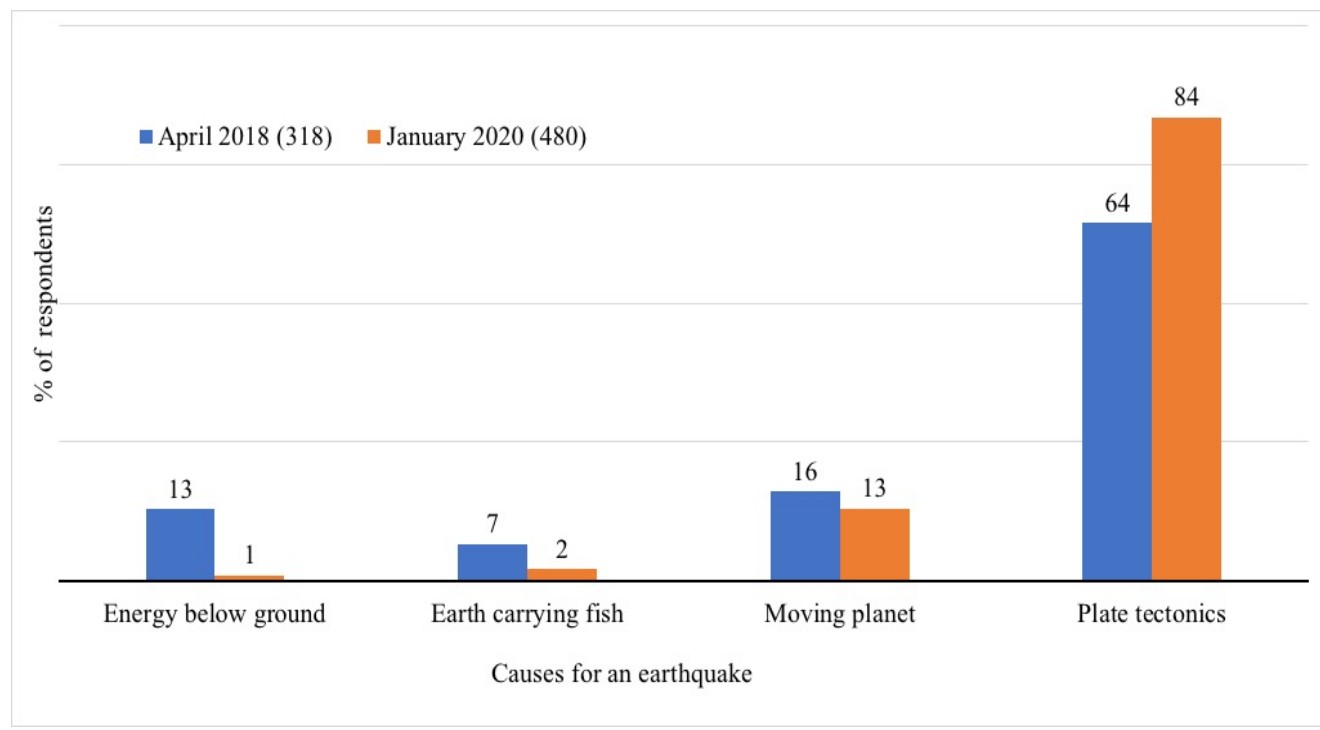



















Figure 6

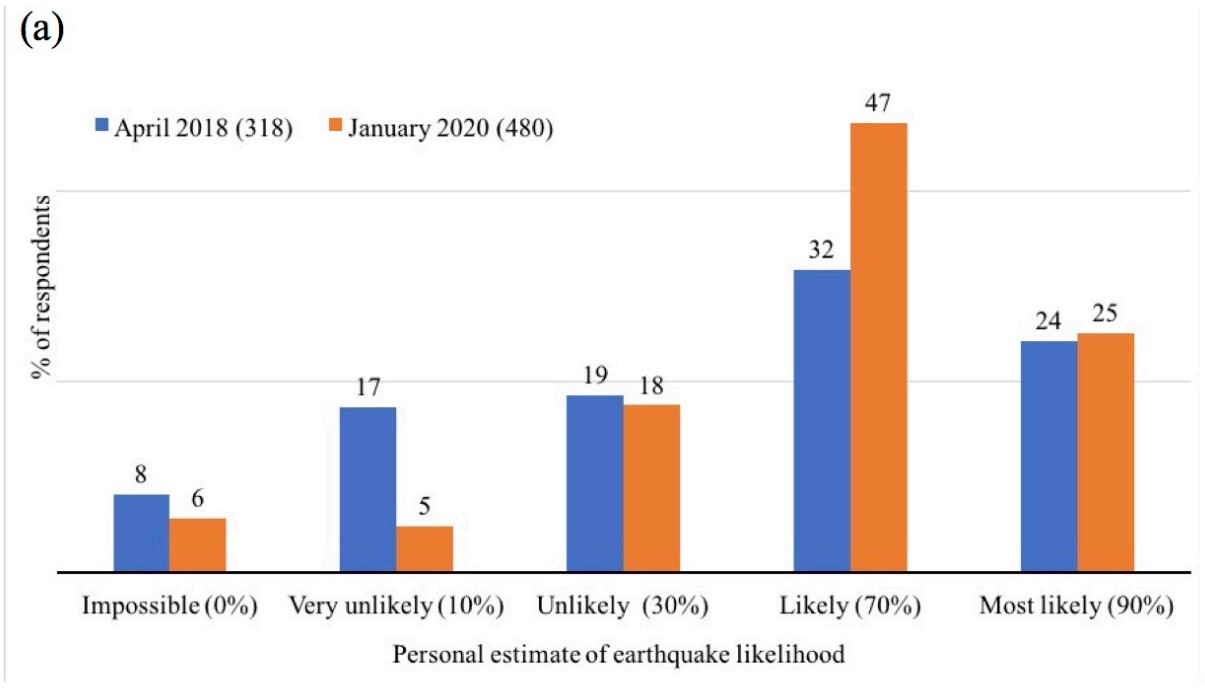

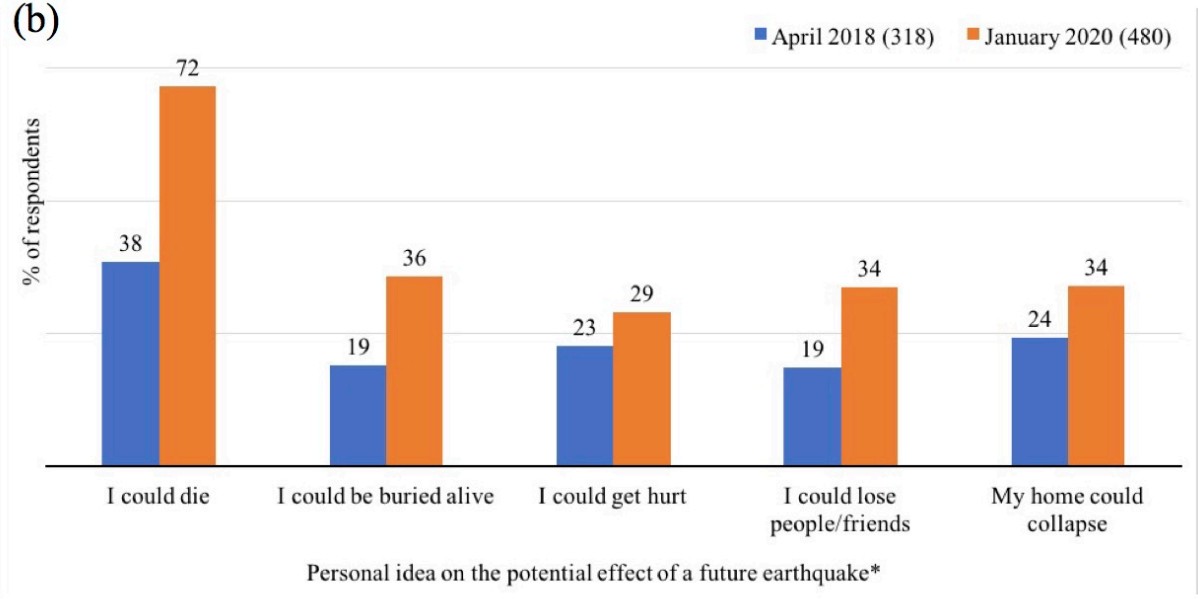








Figure 7

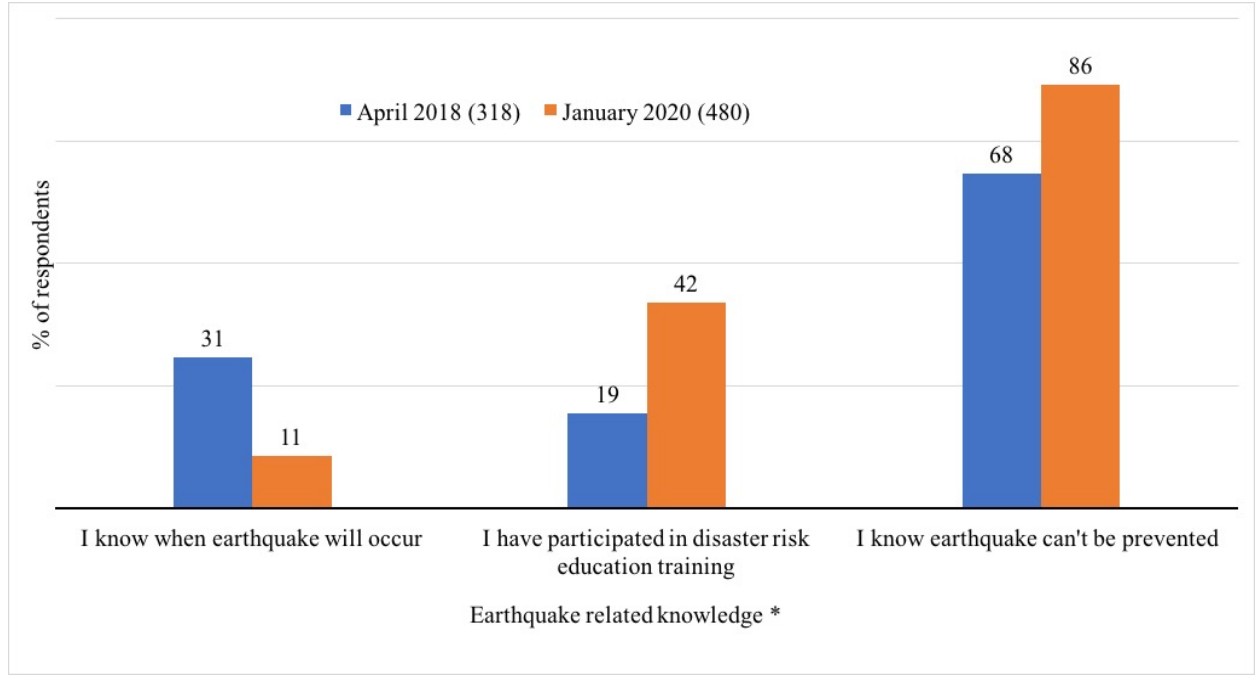



Figure 8

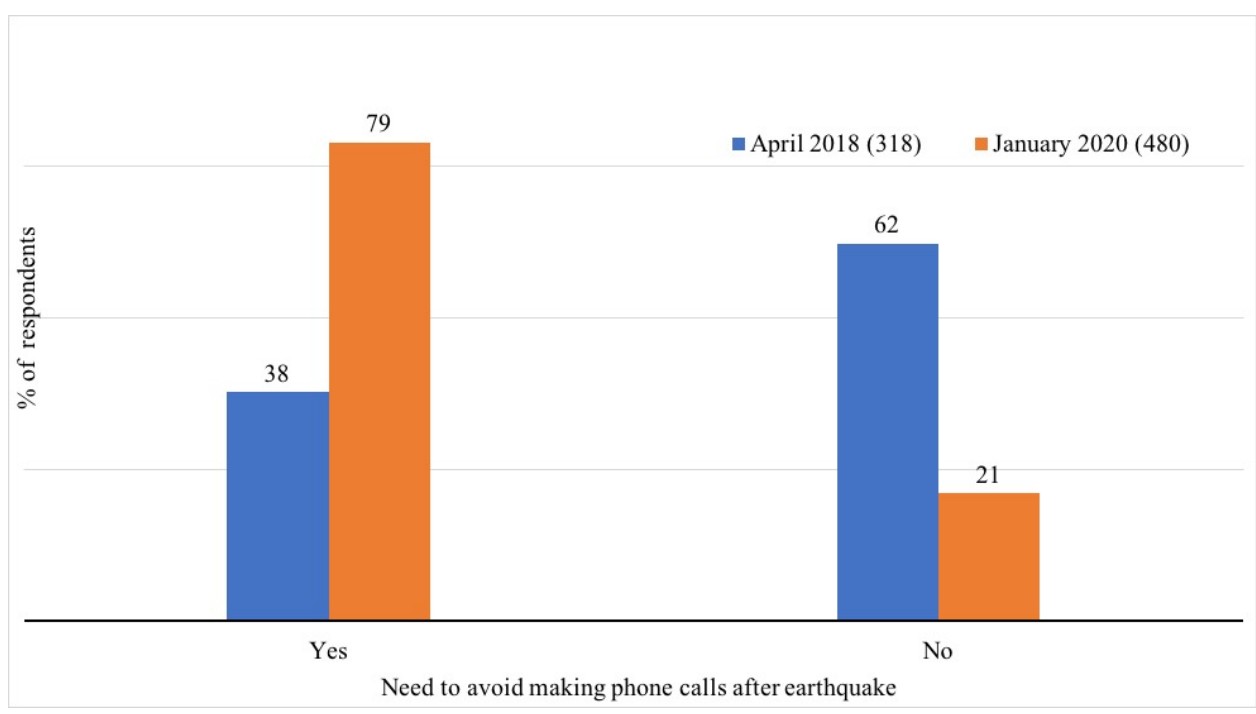



Figure 9

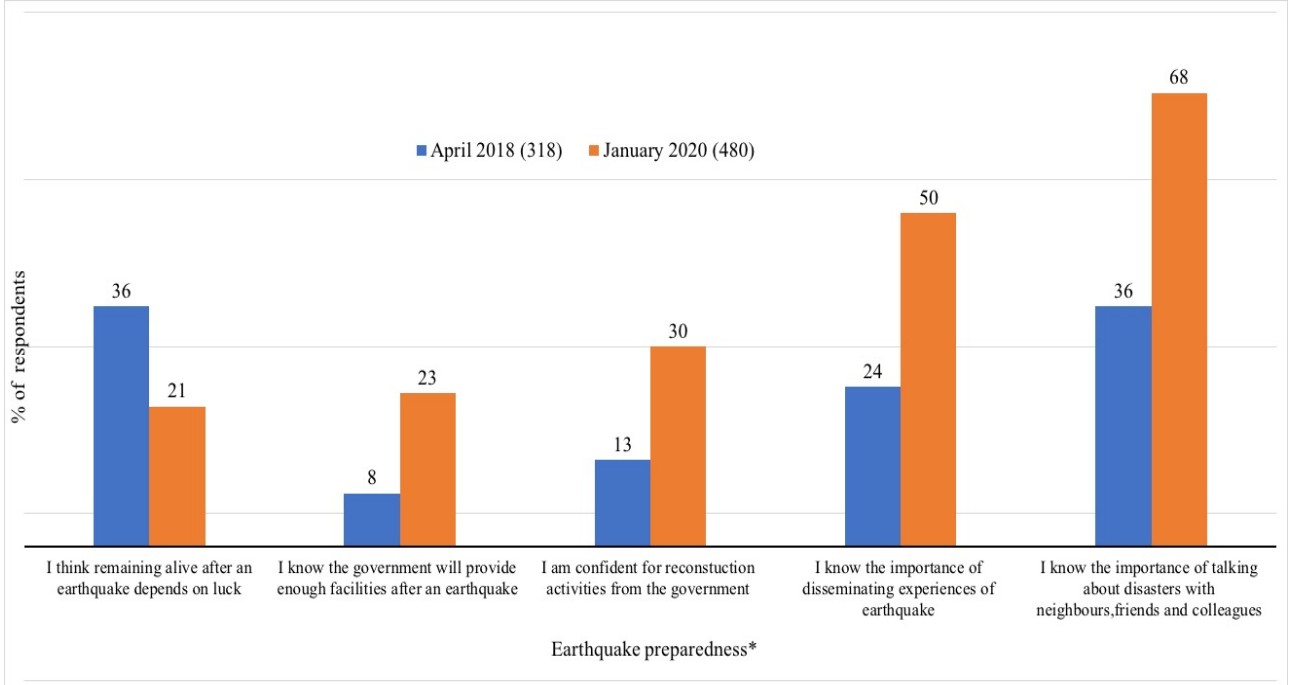




Figure10

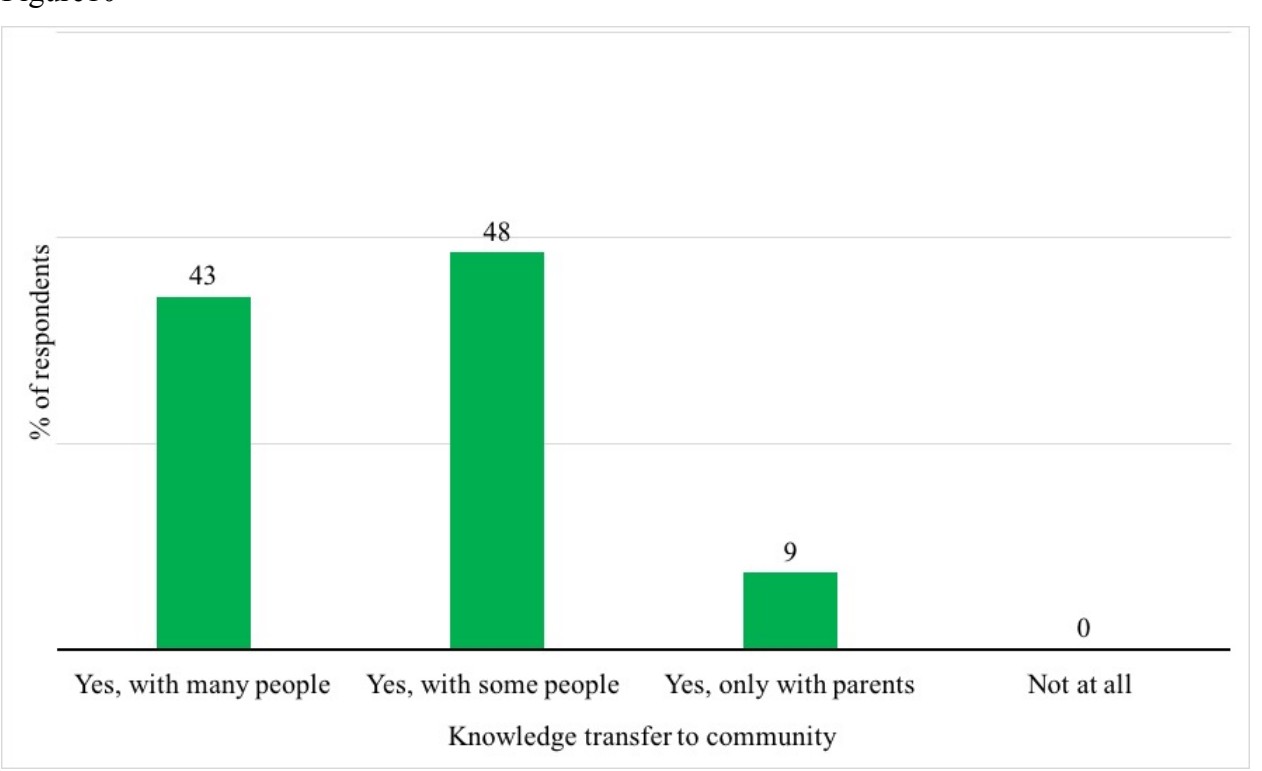



Figure 11

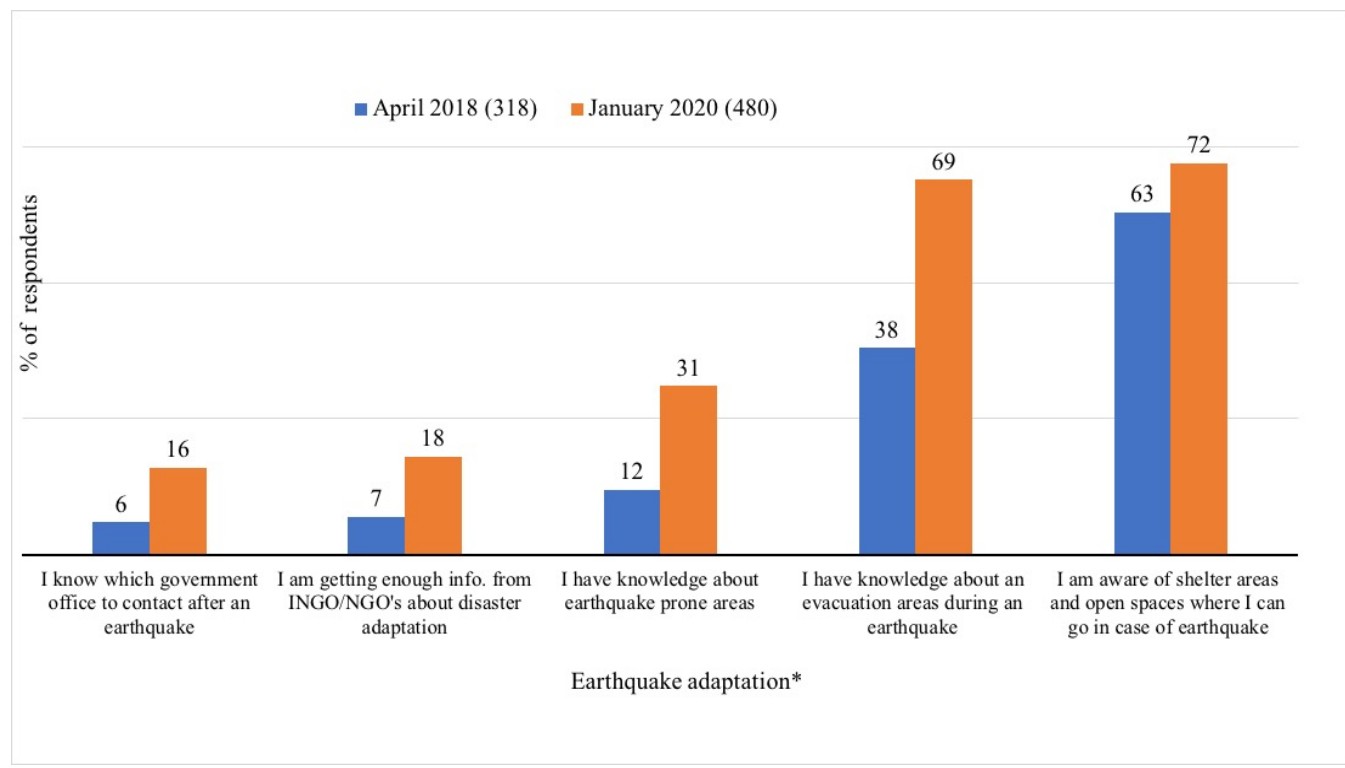




Figure 12

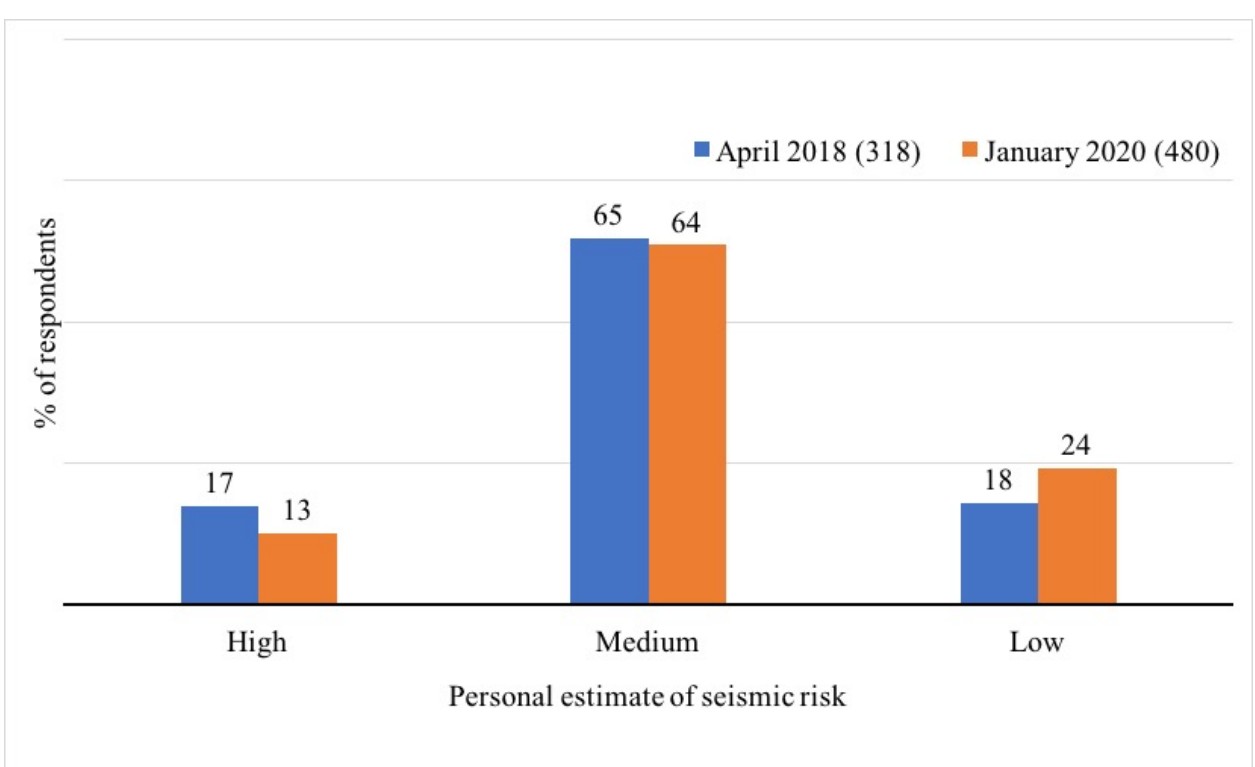
