# Peer review of "Impact of an educational program on earthquake awareness and preparedness in Nepal"

_Geoscience Communication, 2020_

## Referee Comment (RC1) · Michelle Salmon (Referee) · 25 May 2020

I found it refreshing to see an evaluation of this program after it had been running for over a year. Many such programs do not publish such results and it is good to see that there is an impact on the community. I think this provides valuable insight for others that may embark on this kind of program and it certainly shows it's worth. I think the distinction between risk and hazard could perhaps help risk perceptions. I am not sure how much help the statistical calculations are as although they are discussed in the method they are ignored after that. The way in which the percentages are reported can be a little confusing for example on line 161 it says 65% of students believed they could survive in 2020 but then compares this to 57% could NOT survive in 2018. Would this not be better as 43% thought they could survive in 2018 and 65% in 2020.

[Figure]

Despite this the results look clear to me and show that while there is more work to do in preparedness the program is definitely having an impact. It would be good to see a complete questionnaire in the supplementary materials as this could help other programs evaluate their effectiveness in the future (I can't see the question about the seismometer there)

---

## Referee Comment (RC2) · John Taber (Referee) · 31 May 2020

**General comments**

This is a clearly written paper with well justified conclusions based on pre/post surveying of student attitudes, knowledge and behaviors related to earthquake awareness and preparedness. It is impressive to see such a significant effect on student knowledge and their sharing of that increased knowledge with the community after just 2 years. The educational program is very well designed, with a 3-pronged approach of new lessons and lectures taught by the authors, teacher professional development, and the installation of a school seismograph network, all done in collaboration with school leaders. The discussion of the difficulty of changing the perception of risk, even

with the documented increase in earthquake hazard knowledge raised a number of interesting points.

Specific comments

This may be outside the scope of the paper, but I'd be interested to know whether the authors have any data, or can speculate as to the relative importance of the 3 components of the program? Particularly, is it possible to show that having a school seismograph increased student engagement beyond what would have occurred with only lessons and lectures taught by the authors and teacher professional development? If such a claim can be made, it would be helpful for the seismographs in schools community.

Line 143: A little more information would be helpful about the question relating to students knowing when an earthquake will occur, as I didn't quite understand what was being asked. Could you include the full question in the figure 7 caption? That question doesn't seem to be in the supplement.

Figure 8: I suggest modifying the caption to more closely reflect the question, e.g. Need to avoid making phone calls after earthquake.

Technical/typographic corrections

Line 58: missing comma between awareness and preparations.

Line 99: extra "the" : "and the all the"

Line 166: I think this should be "hit by objects, not collapse of constructions"

Line 240: I think this should be "level was increased"

Line 312: I think this should be "implemented and achieved"

Line 327: I think this should be "allowed us to invite"

Table 1, Q11: I think this should be "preparedness for a major"

[Figure]

---

## Referee Comment (RC3) · Lok Bijaya Adhikari (Referee) · 14 Jun 2020

Lok Bijaya Adhikari (Referee)

lbadhikari@hotmail.com

I enjoyed reviewing it which covers the major aspects of Geoscience Communication. I found it scientifically sound and useful for the general public. This is a important work to be done in a country like Nepal which has high seismic hazard. Besides some specific, following comments I don't have major comments for the publication.

Specific Comments Line 54: After or before Fig. 1 mention source/reference. Line 54: After magnitude please mention the type of earthquake before the number. Line 61: It is well-refereed NSET, an NGO working in Nepal however, it is worth mentioning similar activities performed by Government agencies like the National Seismological Center under the Department of Mines and Geology, National Reconstruction Authority, De-

partment of Education, Department of Urban Development and Building Construction, etc. Line 125: Revise the spelling "hid". Line 139: Mention the type of magnitude Line 226: It is better to replace regional and central government by the Government of Nepal only. Line 226: Revise the spelling of "survey". Line 227: Write in the correct order. (eg. Local, Provincial and Federal government) Line 386: It is better to define the term Chi-square, p-value, etc. in the main text. Line 388: Table. Why Q1 to Q6 are not mentioned in the same table? Line 524: Fig. 1 add a table about the location of the school. eg. Lat, Lon, place name, type of school public or privet, number of students, staff, etc. Line 527: Fig. 2 where and which school is this? Line 553: Fig. 4 Correct Nepali word (1. Parba to Purba). In the same figure, some texts are cropped in the bottom row. Line 562: Fig. 5 on-wards, eg. April 2018 (318) and January 2020 (480). As discussed in the main text, the respondents are not repeated from 2018 survey to 2020 survey, it is worth to compare and discuss the variation among the repeated ones.

---

## Author Comment (AC1) · 9 Jul 2020

**Referee 1**

I found it refreshing to see an evaluation of this program after it had been running for over a year. Many such programs do not publish such results and it is good to see that there is an impact on the community. I think this provides valuable insight for others that may embark on this kind of program and it certainly shows it's worth.
Thank you very much.

I think the distinction between risk and hazard could perhaps help risk perceptions. I am not sure how much help the statistical calculations are as although they are discussed in the method they are ignored after that.
Thank you for your suggestion to distinguish between hazard and risk. We will investigate whether the age of the students makes this approach suitable. A repeated survey in 1-2 years may inform us how perceptions change on the mid-term.
Regarding statistical calculations: the Discussion section and the Supplementary Table 1 shows the results. We do not discuss how much change occurred as long as they are significant.

The way in which the percentages are reported can be a little confusing for example on line 161 it says 65% of students believed they could survive in 2020 but then compares this to 57% could NOT survive in 2018. Would this not be better as 43% thought they could survive in 2018 and 65% in 2020.
Yes, it is better to write it that way and we have changed the sentence accordingly.

Despite this the results look clear to me and show that while there is more work to do in preparedness the program is definitely having an impact. It would be good to see a complete questionnaire in the supplementary materials as this could help other programs evaluate their effectiveness in the future (I can't see the question about the seismometer there)
Thank you for this suggestion. We have now inserted the complete questionnaire in the supplementary material.

---

## Author Comment (AC2) · 9 Jul 2020

**Referee 2**

General comments
This is a clearly written paper with well justified conclusions based on pre/post surveying of student attitudes, knowledge and behaviors related to earthquake awareness and preparedness. It is impressive to see such a significant effect on student knowledge and their sharing of that increased knowledge with the community after just 2 years. The educational program is very well designed, with a 3-pronged approach of new lessons and lectures taught by the authors, teacher professional development, and the installation of a school seismograph network, all done in collaboration with school leaders. The discussion of the difficulty of changing the perception of risk, even with the documented increase in earthquake hazard knowledge raised a number of interesting points.
Thank you very much.

Specific comments
This may be outside the scope of the paper, but I'd be interested to know whether the authors have any data, or can speculate as to the relative importance of the 3 components of the program? Particularly, is it possible to show that having a school seismograph increased student engagement beyond what would have occurred with only lessons and lectures taught by the authors and teacher professional development? If such a claim can be made, it would be helpful for the seismographs in schools' community.
Thank you so much, this is a very good idea. We agree that it would be nice to see the impact of the seismometer and special lectures independently and the result could help to develop such educational program in large scale. Unfortunately, there are no schools where a seismometer is not installed and all schools also had special lectures and therefore we do not have a control to test against. We keep this idea in the list of perspectives for the future development of the program.

Line 143: A little more information would be helpful about the question relating to students knowing when an earthquake will occur, as I didn't quite understand what was being asked. Could you include the full question in the figure 7 caption? That question doesn't seem to be in the supplement.
This was a generic question, the title and answer options for this question were as shown on Figure 7, with multiple choice answers. Also, we have now included the complete questionnaire in the supplement, the corresponding question is Q13. We also add each question number at the corresponding figure.

Figure 8: I suggest modifying the caption to more closely reflect the question, e.g. Need to avoid making phone calls after earthquake.
The figure caption was modified with the suggested text.

Technical/typographic corrections
Line 58: missing comma between awareness and preparations.
Thanks, we have inserted the comma.

Line 99: extra "the" : "and the all the"
We have removed the extra 'the' and the sentence now reads "and all the …"

Line 166: I think this should be "hit by objects, not collapse of constructions"

We kept the sentence "hit by objects, collapse of constructions" as most of casualties/injuries are by both of these reasons in developing counties like Nepal, where construction quality is not good enough to resist big earthquakes.

Line 240: I think this should be "level was increased"

We modified the beginning of the sentence and now "have" makes sense.

Line 312: I think this should be "implemented and achieved"

We have added "and" in the sentence.

Line 327: I think this should be "allowed us to invite"

We have changed the sentence accordingly.

Table 1, Q11: I think this should be "preparedness for a major"

We have changed the sentence accordingly.

---

## Author Comment (AC3) · 9 Jul 2020

**Referee 3**

I enjoyed reviewing it which covers the major aspects of Geoscience Communication. I found it scientifically sound and useful for the general public. This is a important work to be done in a country like Nepal which has high seismic hazard. Besides some specific, following comments I don't have major comments for the publication.

Thank you very much.

Specific Comments

Line 54: After or before Fig. 1 mention source/reference.

We have added the reference *Dixit et al., 2013* in the sentence.

Line 54: After magnitude please mention the type of earthquake before the number.

We kept moment magnitude (Mw) in all cases and the sentence is revised.

Line 61: It is well-refereed NSET, an NGO working in Nepal however, it is worth mentioning similar activities performed by Government agencies like the National Seismological Center under the Department of Mines and Geology, National Reconstruction Authority, Department of Education, Department of Urban Development and Building Construction, etc.

The paragraph is updated with this information.

Line 125: Revise the spelling "hid".

We used hid as the past tense of hide.

Line 139: Mention the type of magnitude

In the questionnaire, we have not specified the magnitude type as this complexity is not known to students.

Line 226: It is better to replace regional and central government by the Government of Nepal only.

We have changed the sentence to "Local, Provincial and Federal Government of Nepal" as they have all have some field of possible action.

Line 226: Revise the spelling of "survey".

Sorry for typo, we have corrected it.

Line 227: Write in the correct order. (eg. Local, Provincial and Federal government)

We have changed the sentence accordingly.

Line 386: It is better to define the term Chi-square, p-value, etc. in the main text.

Both parameters are mentioned in the "Methods" section and also in the Statistic sub-section of the main text. For chi-square, we have now inserted the actual Greek symbol.

Line 388: Table. Why Q1 to Q6 are not mentioned in the same table?

Table 1 includes primarily questions related to earthquake preparedness with Yes/No answers, while questions on broader topics with more complex answers are mostly represented in Figures 5-12. In the revised version of the manuscript all questions are listed in the supplementary material.

Line 524: Fig. 1 add a table about the location of the school. eg. Lat, Lon, place name, type of school public or privet, number of students, staff, etc.
This information is already in our earlier publication (Subedi et al. 2020) to which we refer here.

Line 527: Fig. 2 where and which school is this?
This school is Shree Himalaya Secondary School in Barpak, Gorkha district and this information is written in the caption, former Lines number 344-347.

Line 553: Fig. 4 Correct Nepali word (1. Parba to Purba). In the same figure, some texts are cropped in the bottom row.
Thank you for nice catch. We have updated figure with correct word and not-cropped text.

Line 562: Fig. 5 on-wards, eg. April 2018 (318) and January 2020 (480). As discussed in the main text, the respondents are not repeated from 2018 survey to 2020 survey, it is worth to compare and discuss the variation among the repeated ones.
Undoubtedly, it is good to have repeated survey before and after the initiation of the program. As we surveyed high school pupils including grade 10 students, it was almost impossible to repeat the survey with the same persons as grade 10 students normally change school for higher education. For this reason, we can only note that 70 % of schools are the same in both surveys. In addition, we did not ask to fill personal information during the surveys so that students feel more comfortable.

---

## Editor Decision (ED1)

[revised manuscript text omitted]

Figure 2

[Figure]

Figure 3

[Figure]

Figure 4

 भुकम्प जानुपुर्वका तयारी

 भुकम्प जाँदै गर्दा

 भुकम्प पछाडी सतर्क रहनुहोस्

**सुरक्षित ठाउँ पत्ता लगाउनु**

[Figure]

तपाई आफैलाई सुरक्षित गर्नको लागी सुरक्षित ठाउँहरु खोज्नुहोस – टेबलमुनी अथवा बेन्चमुनी वा ढोकाको फ्रेम आदी !

**वरीपरी हेर्नुहोस्**

[Figure]

दराज वा झुण्डाइएका समानहरु राम्रोसँग पर्खालमा(भित्तामा) अडिएको छ छैन चेक गर्नुहोस । अल्लो ठाउँमा भएका गह्रौं सामानहरु हटाउनुहोस । पानीका भाडा, ग्यास चुलो र बिजुलीका स्वीचहरु कहाँ छन् याद गर्नुहोस ।

**अत्यावश्यक सामाग्रीको तयारी**

[Figure]

अत्यावश्यक सामाग्रीको किट (Kit) तयार गर्नुहोस् र सजिले उपलब्ध हुने ठाउँमा राख्नुहोस् ।
अत्यावश्यक सामाग्रीहरु
पानी/ लामो समयसम्म  नकुहिने खानेकुरा / फस्ट एक किट/ सानो ब्याट्री/ टर्चलाइट ब्याट्री सहित/ तातो कपडाहरु/ब्ल्याङ्केट/ आफ्नो परिचय दिने कागजको प्रतिलिपी/ केही पैसा आदी ।

**आफैले अभ्यास गर्नुहोस/ तालीम लिनुहोस**

[Figure]

आधारभुत जीवन रक्षा विधिसँग अभ्यस्त हुनुहोस । यदी परिवारका सदस्यहरु फरक फरक ठाउँमा हुन्छन् भने आपतकालिन अवस्थामा भेट्ने ठाउँ टुङ्गो लगाउनुहोस ।

**कहिले र कहाँ जाने ?**

[Figure]

पहिलो झट्का अनुभव भएको जतिसक्दो छिटो पहिले छनोट गरेको सुरक्षित ठाउँमा जानु  होस । भर्याङ र लिफ्ट प्रयोग नगर्नुहोस । यदी भवनको बाहिर हुनुहुन्छ भने अलिक टाढा जानुहोस ।

**आश्रयस्थल पत्ता लगाउनु**

[Figure]

1. तल जानुहोस
2. आश्रय लिनुहोस
3. बलियोसँग समातुहोस

यदी तपाई विधालयमा हुनुहुन्छ भने तुरुन्तै टेबलमुनी आश्रय लिनुहोस । टेबलका खुट्टाहरु बलियोसँग समातुनुहोस,भुकम्प जाँदै गर्दा टेबलहरु सर्न सक्छन ।

**भवन बाहिरको जोखिम**

[Figure]

यदी तपाई भवनबाहिर हुनुहुन्छ भने बाहिरै बस्नुहोस र भवनभन्दा टाढा जानुहोस, उच्च विद्युतिय लाइन वा अन्य वस्तुहरु खस्न सक्ने ठाउँभन्दा टाढा जानुहोस । भिरालो ठाउँबाट टाढा जानुहोस भुकम्पले पहिरो जान सक्छ, ढुङ्गा खस्न सक्छन ।

**कारभित्र/बसभित्र**

[Figure]

यदी तपाई कार/बसभित्र हुनुहुन्छ भने खुल्ला ठाउँमा रोक्नुहोसर सवारी भित्र नै बस्नुहोस । पुलमाथी तथा आकाशे पुलमुनी पार्किङ नगर्नुहोस ।

**भुकम्पको झड्का सकिदा बित्तिकै**

[Figure]

जब पहिलो झड्का रोकिन्छ, यदी सम्भव छ भने पानी/ग्यास र बिजुली बन्द गर्नुहोस । अत्यावश्यक सामाग्री लिनुहोस र भवनबाट बाहिर निस्कनुहोस् ।

**सावधानीपूर्वक बस्नुहोस**

[Figure]

चेतावनी ! एउटा भुकम्प पछाडी अरु कम्पनहरु पनि आउन सक्छन, जसलाई पराकम्पन भनिन्छ ।
कम्पनको कारणले उत्पन्न हुने अन्य जोखिमहरु जस्तै पहिरो, बाढी, आगोलागीको बारेमा सचेत हुनुहोस ।

**मेडिकल केयरको सुनिश्चित गर्नुहोस।**

[Figure]

तपाई आफ्नो चोट जाँच गर्नुहोस ,तपाईको वरिपरी यदी कोही जटिल र अप्ठ्यारो अवस्थामा देख्नुभयो भने सहयोग गर्नुहोस ।
अन्य जानकारीका लागी र निर्देशनहरुका लागी रेडियो/टेलीभिजन सुन्नुहोस ।

**अत्यावश्यक सेवाहरु**

[Figure]

आफ्ना साथी र परिवारलाई बारम्बार सम्पर्क गरेर टेलीफोन लाइन व्यस्त नबनाउँनुहोस । अत्यावश्यक सेवालाई प्राथमिकता दिनुहोस ।

Figure 5

[Figure]

Figure 6

[Figure]

[Figure]

Figure 7

[Figure]

Figure 8

[Figure]

Figure 9

[Figure]

Figure10

[Figure]

Figure 11

[Figure]

Figure 12

[Figure]

---

## Author Response (AR2)

[revised manuscript text omitted]

Figure 2

[Figure]

Figure 3

[Figure]

Figure 4

[Figure]

**सुरक्षित ठाउँ पत्ता लगाउनु**

[Figure]

तपाई आफैलाई सुरक्षित गर्नको लागी सुरक्षित ठाउँहरू खोज्नुहोस - टेबलमुनी अथवा बेन्चमुनी वा ढोकाको फ्रेम आदी !

**वरीपरी हेर्नुहोस्**

[Figure]

दराज वा झुण्डाइएका समानहरू राम्रोसँग पर्खाल्सँग(भित्तामा) अड्डिएको छ छैन चेक गर्नुहोस । अल्गो ठाउँमा भएका गह्रौं सामानहरू हटाउनुहोस । पानीका भाडा, ग्यास चुलो र बिजुलीका स्विचहरू कहाँ छन् याद गर्नुहोस ।

**अत्यावश्यक सामाग्रीको तयारी**

[Figure]

अत्यावश्यक सामाग्रीको किट (Kit) तयार गर्नुहोस र सजिलै उपलब्ध हुने ठाउँमा राख्नुहोस ।
अत्यावश्यक सामाग्रीहरू
पानी/ लामो समयसम्म  नकुहिने खानेकुरा / फस्ट एक किट/ सानो ब्याट्री/ टर्चलाइट ब्याट्री सहित/ तातो कपड़ाहरू/ब्ल्याङ्केट/ आफ्नो परिचय दिने कागजको प्रतिलिपी/ केही पैसा आदी ।

**आफैले अभ्यास गर्नुहोस्/ तालीम लिनुहोस्**

आधारभूत जीवन रक्षा बिधिसँग अभ्यस्त हुनुहोस । यदी परिवाका सदस्यहरू फरक फरक ठाउँमा हुन्छ भने आपतकालिन अवस्थामा भेट्ने ठाउँ ढुङ्गो लगाउनुहोस ।
* * *
[Figure]

**कहिले र कहाँ जाने ?**

[Figure]

पहिलो झट्का अनुभव भएको जतिसक्दो छिटो पहिलै छनोट गरेको सुरक्षित ठाउँमा जानु  होस । भर्याङ र लिफ्ट प्रयोग नगर्नुहोस । यदी भवनको बाहिर हुन्छ भने अलिक टाढा जानुहोस ।

**आश्रयस्थल पत्ता लगाउँनु**

[Figure]

यदी तपाई विधालयमा हुनुहुन्छ भने तुरुन्तै टेबलमुनी आश्रय लिनुहोस । टेबलका खुट्टाहरू बलियोसँग समात्नुहोस,भुकम्प जाँदै गर्दा टेबलहरू सर्न सक्छन ।

**भवन बाहिरको जोखिम**

[Figure]

यदी तपाई भवनबाहिर हुनुहुन्छ भने बाहिरै बस्नुहोस र भवनभन्दा टाढा जानुहोस, उच्च विद्युतिय लाइन र अन्य वस्तुहरू खस्न सक्ने ठाउँभन्दा टाढा जानुहोस । भिरालो ठाउँबाट टाढा जानुहोस भुकम्पले पहिरो जान सक्छ, दुङ्ग खस्न सक्छन ।

**कारभित्र/बसभित्र**

[Figure]

यदी तपाई कार/बसभित्र हुनुहुन्छ भने खुल्ला ठाउँमा रोक्नुहोस,सवारी भित्र नै बस्नुहोस । पुलमाथी तथा आकाशे पुलमुनी पार्किङ नगर्नुहोस ।
* * *
[Figure]

**भुकम्पको झड्का सकिदा बित्तिकै**

[Figure]

जब पहिलो झड्का रोकिन्छ, यदी सम्भय छ भने पानी/ग्यास र बिजुली बन्द गर्नुहोस । अत्यावश्यक सामाग्री लिनुहोस र भवनबाट बाहिर निस्कनुहोस ।

**सावधानीपूर्वक बस्नुहोस**

[Figure]

चेतावनी । एउटा भुकम्प पछाडी अरू कम्पनहरू पनि आउँन सक्छन, जसलाई पराकम्पन भनिन्छ । कम्पनको कारणले उत्पन्न हुने अन्य जोखिमहरू जस्तै पहिरो, बाढी, आगोलागीको बारेमा सचेत हुनुहोस ।

**मेडिकल केयरको सुनिश्चित गर्नुहोस्।**

[Figure]

तपाई आफ्नो चोट जाँच गर्नुहोस, तपाईको वरिपरी यदी कोही जटिल र अप्ठ्यारो अवस्थामा देख्नुभयो भने सहयोग गर्नुहोस । अन्य जानकारीका लागी र निर्देशनहरूका लागी रेडियो/टेलिभिजन सुन्नुहोस ।

**अत्यावश्यक सेवाहरू**

[Figure]

आफ्ना साथी र परिवारलाई बारम्बार सम्पर्क गरेर टेलेफोन लाइन व्यस्त नबनाउँनुहोस । अत्यावश्यक सेवालाई प्राथमिकता दिनुहोस ।

Figure 5

[Figure]

Figure 6

(a)

[Figure]

(b)

[Figure]

Figure 7

[Figure]

Figure 8

[Figure]

Figure 9

[Figure]

Figure10

[Figure]

Figure 11

[Figure]

Figure 12

[Figure]